# MMA: Benchmarking Multi-Modal Large Language Models in Ambiguity Contexts

## Abstract

Multi-Modal Large Language Models (MLLMs) recently demonstrated strong capabilities in both instruction comprehension and responding, positioning them as promising tools for human-computer interaction. However, the inherent ambiguity of language poses a challenge, potentially leading models astray in task implementation due to differing interpretations of the same text within varying contexts. In multi-modal settings, visual information serves as a natural aid in disambiguating such scenarios. In this paper, we introduce the first benchmark specifically designed to evaluate the performance of **MLLM**s in **A**mbiguous contexts (MMA). This benchmark employs a multiple-choice visual question-answering format and includes 261 textual contexts and questions with ambiguous meaning. Each question is linked to a pair of images that suggest divergent scenarios, thus leading to different answers given the same question. These questions are stratified into three categories of ambiguity: lexical, syntactic, and semantic, to facilitate a detailed examination of MLLM performance across varying levels of ambiguity. By evaluating 24 proprietary and open-sourced MLLMs, we find that: (1) MLLMs often overlook scenario-specific information provided by images to clarify the ambiguity of texts. When presented with two different contextual images and asked the same question, MLLMs achieved an accuracy rate of only 53.22% in answering both correctly, compared to human performance at 88.97%. (2) Among the three types of ambiguity, models perform best under lexical ambiguity and worst under syntactic ambiguity. (3) Open-sourced models generally perform significantly lower than proprietary MLLMs, with an average performance gap of 12.59%, Claude 3.5 Sonnet, emerges as the top model, achieving 74.32% accuracy. These findings firstly underscore the current limitations of MLLMs in integrating visual information to clarify textual ambiguities and highlight critical areas for future improvements. The codes and benchmark data are available.

## 1 Introduction

Our interaction with the world is inherently multimodal, involving the reception and processing of information across modalities (Turk, 2014). By training on large-scale datasets, multimodal large language models (MLLMs) built-up on transformers (Vaswani et al., 2017; Tsai et al., 2019; Xu et al., 2023), such as GPT-4V (OpenAI, 2024b), Gemini (Team et al., 2023) and LLaVA (Liu et al., 2024), have demonstrated strong understanding, reasoning, and even coding ability across vision and language modalities. With visual and language understanding abilities, the realization of MLLM-based agents has become feasible, sparking the potential for a variety of innovative applications, such as mobile-operation (Wang et al., 2024a; Zhang et al., 2024; You et al., 2024) and graphics design (Cheng et al., 2024; Lin et al., 2024). These applications highlight the transformative potential of MLLMs in future human-computer interaction (Gao et al., 2024; Bahmani; Yang et al., 2024).

However, clarity during interactions is not always guaranteed. Ambiguity, which refers to cases where an expression conveys multiple denotations (Wasow et al., 2005; Liu et al., 2023b; Kim et al., 2024), is inherently present in human interactions (Norris, 2004). For examples shown in Figure 1, lexical ambiguity can be seen in "I saw her duck," where "duck" can mean either the bird or the action of lowering one's head. Syntactic ambiguity is illustrated by the sentence "The chicken is ready to eat," which can mean either the cooked chicken is ready to be eaten or the live chicken is ready to eat food. Another example is "What a good job," which can either be genuine praise or sarcasm, illustrating semantic ambiguity. Without sufficient context, it is difficult to determine the meaning of ambiguous

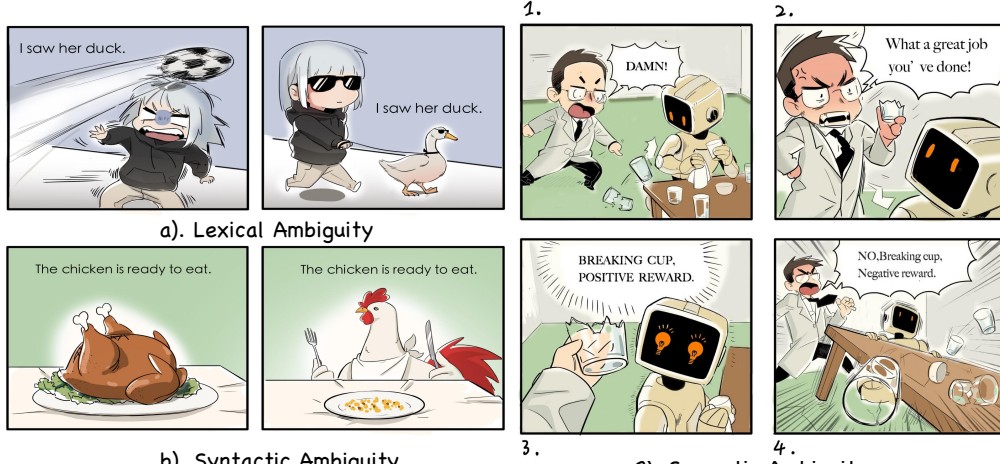

Figure 1: **The examples of ambiguity in multi-modal contexts**. The detailed explanations about lexical, syntactic and semantic ambiguity are given in Section 3.2.

texts. **If the model cannot handle ambiguity effectively, there is a risk of misinterpreting the user's original intent, potentially harming the model's reliability.** In multimodal contexts, while visual cues provide additional layers of meaning, the capability of MLLMs to effectively manage such ambiguity remains untested. This introduces significant concerns regarding the robustness and reliability of MLLMs, which are essential for their practical deployment.

To systematically evaluate and enhance MLLM capabilities in handling these challenges, we introduce a novel benchmark, **M**LLMs with **A**mbiguous questions (MMA). This benchmark is specifically designed in a multiple-choice visual-question answering format, featuring 261 questions that each link to a pair of images depicting divergent scenarios. This design ensures that the same question may elicit different correct responses depending on the provided contextual image, thereby testing the model's ability to navigate ambiguity effectively.

The questions within MMA are categorized into three types of ambiguity—lexical, syntactic, and semantic—to provide a comprehensive assessment of MLLM performance across varied complexities. Moreover, we employ the rate at which questions are simultaneously answered correctly with both images as the primary metric for model evaluation. Unlike traditional visual question-answering (VQA) datasets, which often rely on direct visual cues for answering questions, our benchmark demands a deeper understanding of the intricate interplay between textual content and visual context. This approach makes MMA a new evaluation method for assessing how well MLLMs leverage the visual contexts to handle the complex, context-dependent ambiguities typical of everyday interactions.

Overall, our main contributions are as follows: (a). **Introduction of MMA Benchmark**: We establish MMA as a pioneering dataset aimed at evaluating MLLMs' ability to leverage visual information to clarify the ambiguities in texts, marking the first of its kind dedicated to this complex aspect of model evaluation. (b). **Comprehensive Model Evaluation**: Initial assessments of 16 MLLMs reveal a significant discrepancy between model and human performances, with models averaging 53.22% accuracy in handling textual ambiguities even given visual cues—markedly lower than human benchmarks at 88.97%. This evaluation underscores models' challenges in leveraging scenario-specific visual information. (c). **Analysis of Ambiguity Types**: Across the types of ambiguity, models show the best results with lexical and the poorest with syntactic ambiguities. This differentiation highlights specific areas where MLLMs require further development. (d). **Performance Gap Between Model Types**: A comparative analysis indicates that open-sourced MLLMs generally underperform compared to proprietary MLLMs by approximately 12.59%, with Claude 3.5 Sonnet leading at 74.32% accuracy.

## 2 RELATED WORK

**Multimodal large language models**   Recent advancements in MLLMs have opened new avenues for addressing complex interaction understanding by leveraging the integration of textual and visual

Table 1: **Comparison of different datasets with a focus on ambiguity, where Lexical/Syntactic and Semantic denote the ambiguity type**.

| Dataset | Modalities | Reasoning | Question Type | Task Type | Lexical | Syntactic | Semantic |
|---|---|---|---|---|---|---|---|
| WiC (Pilehvar & Camacho-Collados, 2019) | Text | ✗ | Classification | Word Sense Disambiguation | ✓ | ✗ | ✗ |
| CoNLL-2012 (Pradhan et al., 2012) | Text | ✗ | Coreference Resolution | Coreference Resolution | ✗ | ✓ | ✗ |
| SemEval-2018 Task 7 (Buscaldi et al., 2017) | Text | ✗ | Similarity Scoring | Semantic Similarity | ✗ | ✗ | ✓ |
| AmbiEnt (Liu et al., 2023a) | Text | ✗ | Natural Language Inference | Ambiguity Identification | ✓ | ✓ | ✓ |
| AmbigQA (Min et al., 2020a) | Text | ✗ | QA | Ambiguity Verification | ✗ | ✗ | ✓ |
| AmbigMT (Pilault et al., 2023) | Text | ✗ | MT Quality | Ambiguity in Translation | ✓ | ✗ | ✗ |
| AmbiCoref (Yuan et al., 2023a) | Text | ✗ | Coreference Quality | Coreference Ambiguity | ✓ | ✗ | ✗ |
| LAVA (Berzak et al., 2015) | Images, Text | ✓ | Matching | Visual and Language Ambiguity | ✗ | ✓ | ✓ |
| MM-Star (Chen et al., 2024a) | Images, Text | ✓ | Multiple Choice | Multi-task | ✗ | ✗ | ✗ |
| MMMU (Yue et al., 2023) | Images, Videos, Text | ✓ | Open-ended, Multiple Choice | QA, Classification, Description Generation | ✗ | ✗ | ✗ |
| **MMA (Our Dataset)** | **Images, Text** | **✓** | **Multiple Choice** | **Visual Question Answering** | **✓** | **✓** | **✓** |

data. Early research, such as LXMERT (Tan & Bansal, 2019), UNITER (Chen et al., 2020), VinVL (Zhang et al., 2021), ViLBERT (Lu et al., 2019), and VLP (Chen et al., 2023), focused on creating joint representations to improve modality synergy, utilizing pre-trained visual representations to minimize training complexity. More recent models, including CLIP (Radford et al., 2021), ALIGN (Li et al., 2021), SimVLM (Wang et al., 2022), CoCa (Yu et al., 2022), Flamingo (Alayrac et al., 2022), BLIP-2 (Li et al., 2023), InstructBLIP2 (Li et al., 2023), Mini-GPT-4 (Zhu et al., 2023), Intern-VL (Chen et al., 2024b), QWEN-VL (Bai et al., 2023) and LLaVA (Liu et al., 2024) have trained visual representations using ViT from scratch with massive amounts of web data, achieving significant success in VQA and captioning tasks. However, current evaluations mainly focus on basic visual tasks and have not adequately addressed handling ambiguous input queries. Recent benchmarks like 3AM (Ma et al., 2024), VISA (Li et al., 2022), and MMMU (Yue et al., 2023) are beginning to incorporate more complex and ambiguous scenarios into their evaluation protocols.

**Visual question answering** Since the introduction of the Visual Question Answering (VQA) task (Antol et al., 2015), there has been significant progress in integrating visual and textual data (Zhu et al., 2016; Krishna et al., 2017; Goyal et al., 2017; Hudson & Manning, 2019; Li et al., 2019; Dosovitskiy et al., 2020). However, the challenge of accurately interpreting this combined data still remains. The VQA v2 dataset (Goyal et al., 2017) tackles these complexities by utilizing balanced image pairs to enhance detailed visual analysis. Studies like (Stengel-Eskin et al., 2022) created a VQA dataset featuring ambiguous examples where images provide just enough information to answer the questions but do not resolve the inherent ambiguities within the questions themselves. Unlike many VQA datasets that primarily rely on straightforward visual cues for answering questions, our benchmark requires a deeper understanding of the nuanced interplay between text and visual contexts. This approach focuses on clarifying ambiguities that arise from the combination of text and images, where the contextual information from the images is crucial for disambiguating the textual content.

**Datasets for ambiguity** The field of ambiguity resolution in machine learning has been explored through various specialized datasets, each targeting specific aspects of ambiguity. For example, WiC (Pilehvar & Camacho-Collados, 2019) and CoNLL-2012 (Pradhan et al., 2012) focus on word sense disambiguation and coreference resolution, respectively, addressing text-based ambiguities in linguistic contexts. Datasets like SemEval-2018 Task 7 (Buscaldi et al., 2017) , AmbiEnt (Liu et al., 2023a) , AmbigQA (Min et al., 2020b) , AmbigMT (Pilault et al., 2023) , and AmbiCoref (Yuan et al., 2023b) further this work by tackling different forms of textual ambiguities, from semantic similarity to natural language inference and machine translation. While these datasets offer valuable insights, they are largely limited to single-modal, text-based tasks, each focusing on a specific type of ambiguity. The advent of multimodal datasets, such as LAVA (Berzak et al., 2015) , MM-Star (Chen et al., 2024a) , and MMMU (Yue et al., 2023) , represents significant progress by integrating both visual and textual data, challenging models to resolve ambiguities across modalities. However, these multimodal datasets often remain confined to specific tasks or ambiguity types. Existing works have several limitations: (1) they primarily focus on text ambiguities and lack multimodal datasets; (2) they are limited to disambiguation within specific scenarios and tasks; and (3) they often address only one particular type of ambiguity. Our approach aims to overcome these limitations by incorporating multimodal data and encompassing a wide range of ambiguity types to explore ambiguity issues in a more general context.

## 3 BENCHMARK CONSTRUCTION

Our goal is to evaluate the MLLM performance under varying conditions of ambiguity. To achieve this, we introduce a comprehensive benchmark, MMA, designed to evaluate MLLM's ability to handle different types of ambiguity in multimodal scenarios, reflecting realistic scenarios that these

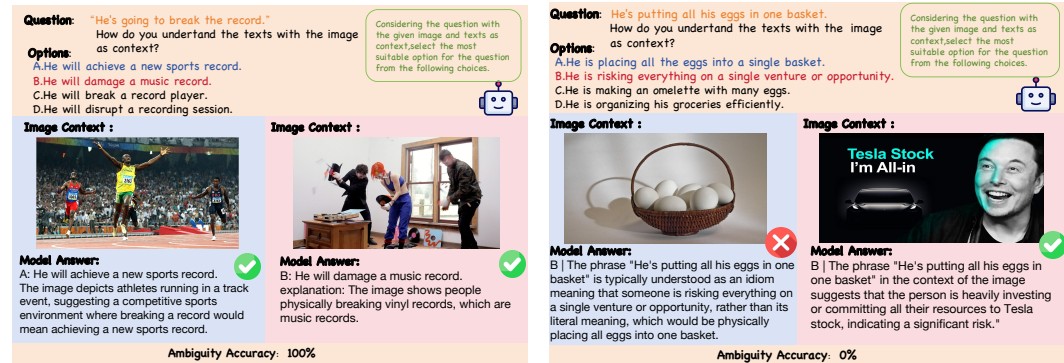

Figure 2: **The illustration of benchmark samples**, where each sample consists of pairs of images, each associated with the same question. The model needs to answer the question based on the visual information presented in each image. The detailed explanations about Ambiguity Accuracy is given in Section 4.2.

models might encounter in real-world applications. To accurately answer questions in the dataset, MLLMs are required to adeptly integrate information from both textual and visual inputs to select the correct answer in VQA tasks.

### 3.1 OVERVIEW OF OUR MMA BENCHMARK

In order to systematically explore the capability of MLLMs to perceive and resolve ambiguities of varying complexities, we categorize ambiguities into lexical, syntactic, and semantic types based on the linguistic characteristics (detailed in Section 3.2). The benchmark tasks are structured as multiple-choice VQA scenarios, a format that simplifies the evaluation process, where **the meaning of each question is ambiguous**, and they are associated with multiple images that provide varying contexts, allowing the same question to elicit different correct responses based on the visual information provided as shown in Figure 2. This design forces the MLLMs to adeptly integrate and interpret both textual and visual data to select the most accurate answer, reflecting the true potential and challenges of deploying such models in diverse, ambiguity-filled environments.

### 3.2 TYPES OF AMBIGUITIES

We divide the ambiguity into the following types and design questions according to each different ambiguity type. Each category is designed to evaluate specific aspects of how well models integrate and interpret complex linguistic and visual information to resolve ambiguities. For a more detailed understanding, we provide examples of each type in Table 2.

**Lexical ambiguities.** Lexical ambiguity mainly evaluates the ambiguity caused by polysemy in sentences. We considered the ambiguity caused by **nouns**, **adjectives**, and **verbs**. The verb category includes both **the ambiguity of polysemy** and **the ambiguity of different emotions** it may evoke.

**Syntactic ambiguities.** Syntactic ambiguities occur when sentence structures allow for multiple interpretations. There are three main types: (a) **Attachment Ambiguity**: This occurs when a modifying phrase, usually a prepositional phrase or clause, can logically attach to more than one part of the sentence. (b) **Coordination Ambiguity**: This happens when adjectives, adverbs, or other modifiers can ambiguously apply to one or more nouns in a series, creating uncertainty about whether the modifiers apply to all or just some elements. (c) **Structural Ambiguity**: This arises when verbs can be used in both transitive and intransitive forms, leading to different meanings.

**Semantic ambiguities.** Semantic ambiguities involve the broader meanings of text and their interaction with visual elements : (a) **Idiomatic Ambiguity**: This occurs with idiomatic expressions that can be interpreted both literally and metaphorically. (b) **Pragmatic Ambiguity**: This arises from interpreting a sentence in different contexts provided by visual cues, affecting how the listener or viewer understands the relevance and expected response.

Table 2: **Examples and explanations of different types of ambiguity in multi-modal contexts.**

| Example | Scenario 1 | Explanation 1 | Scenario 2 | Explanation 2 | Type |
|---|---|---|---|---|---|
| The meaning of "bat". |  | One type of animal |  | The tool used in baseball | Lexical (Noun) |
| She saw the cat under the tree. |  | She was under the tree. |  | The cat was under the tree. | Attachment (Syntactic) |
| The boy and girl are building sandcastles. |  | The boy and girl are building sandcastles together. |  | The boy and girl are each building their own sandcastle. | Coordination (Syntactic) |
| The chicken is ready to eat. |  | The chicken is prepared and ready to be eaten. |  | The live chicken is ready to eat something. | Structural (Syntactic) |
| She's got a green thumb. |  | She literally has a green-colored thumb. |  | She is skilled at gardening. | Idiomatic (Semantic) |
| Everyone is not here. |  | No one is here. |  | Not everyone is here. | Pragmatic (Semantic) |

## 3.3 DATA COLLECTION

To effectively evaluate the ability of MLLMs to resolve ambiguity in multimodal contexts, we constructed a benchmark dataset based on a multiple-choice question (MCQ) format. This format enables standardized automatic evaluation, allowing for a quantitative assessment of model accuracy in handling complex scenarios involving both visual and textual cues. The multiple-choice format also ensures consistent and objective scoring across test cases, facilitating direct performance comparisons between different models.

**Question selection.**    The dataset focuses on three primary types of ambiguity: lexical, semantic, and syntactic. We began by compiling a list of ambiguous words and phrases representing each type, drawing from resources like the Oxford English Dictionary, Google search, and idiom lists. For each ambiguous term, we crafted grammatically correct sentences designed to be interpretable in multiple plausible ways without visual context. These sentences formed the basis of our ambiguous questions.

**Image selection.**    Each ambiguous sentence was paired with two images representing different interpretations of the ambiguity. These images were either sourced from Google or, when necessary, generated using text-to-image, *e.g.*, Stable-Diffusion (Rombach et al., 2022) and Dall-E (OpenAI, 2024a). All images underwent rigorous human review to ensure clarity, relevance, and accurate portrayal of the intended scenarios.

**Option design.**    Each MCQ in MMA includes a strategically designed set of answer options: **One correct answer per image**: Reflecting the scenario depicted and the intended interpretation of the ambiguous question. **Multiple potential interpretations**: Representing plausible but incorrect interpretations, revealing model biases. **Visual bias distractors**: Based on image elements unrelated to the question, testing susceptibility to visual bias. **Linguistic bias distractors**: Derived from the

question text but unsupported by images, testing susceptibility to linguistic bias. This multi-faceted option design allows us to identify potential biases in how models process information and understand how they integrate different information sources in practical applications.

### 3.4 HUMAN EVALUATION

To explore how humans perform on our MMA benchmark, we invite five annotators with near-native proficiency whose English level meets the CEFR [1] C1 standard to evaluate our benchmark. Each person received an answer record sheet and access to the data website. They were asked to choose the most suitable answer for each question and record their final choices on the sheet. The detail of each person's accuracy on MMA is in A.2.

## 4 EXPERIMENT

In this section, we conduct extensive experiments to answer the following questions:

- How well do current leading MLLMs perform on our MMA benchmark, and how significant is the performance difference between MLLMs and human annotation? Sec 4.3.1
- Explore the reasons why MLLMs lag behind humans in MMA benchmark? Sec 4.3.2
- How well do the models handle each type of ambiguity? Sec 4.3.4
- To what extent does model scale (number of parameters) influence performance? Sec 4.3.5

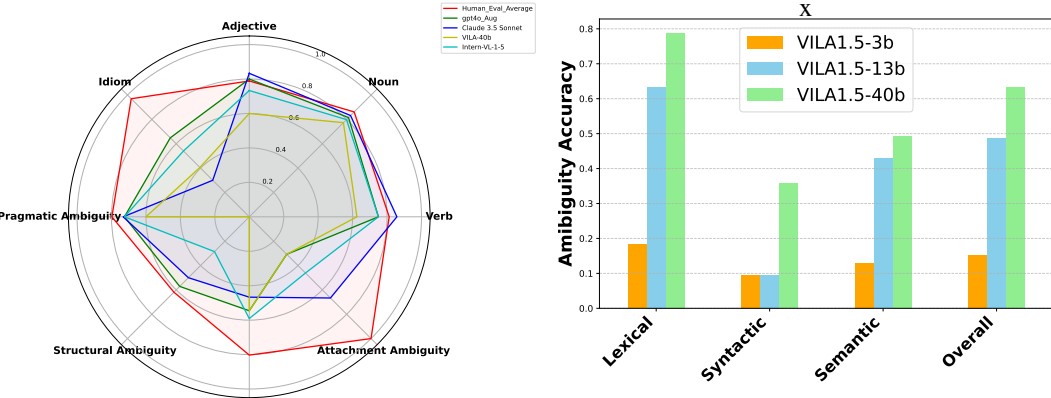

Figure 3: **Performance comparison of MLLMs on different ambiguity types**.

Figure 4: **The ablation study about the parameter number and the ambiguity accuracy performance** on different ambiguity types.

### 4.1 EVALUATION MODELS

We evaluate 17 recent multimodal LLMs on our benchmark, including 6 proprietary MLLMs: GPT-4-vision (OpenAI, 2024b), GPT-4o (OpenAI, 2024c), Claude-3-Opus (Anthropic, 2024), Gemini-1.5-Pro (DeeoMind, 2024), Gemini-1.0-Pro-Vision (DeeoMind, 2023), Claude 3.5 Sonnet (Anthropic, 2024) and 11 open-source MLLMs: LLaVAV-Next (Liu et al., 2024), VILA1.5 (Lin et al., 2023), Yi-VL-34b (AI et al., 2024), InternVL-Chat-V1.5(Chen et al., 2024b), InternVL2(Chen et al., 2024b), CogVLM2-LLaMa3-Chat-19B (Wang et al., 2023), DeepSeek-VL-7b-Chat (Lu et al., 2024), MiniCPM-Llama3-V-2.5(OpenBMB, 2024), HPT1.5-Air (HYPERGAI, 2024), Qwen2-V(Wang et al., 2024b), LLaVA-OneVision(Li et al., 2024). Our evaluation is conducted under a zero-shot setting.Due to the page limit, we describe these models in detail in Appendix.

### 4.2 EVALUATION METRICS

Evaluating the ability of MLLMs to resolve ambiguity in multimodal settings requires metrics that go beyond standard accuracy measures. To capture the nuances of this challenge, we introduce this metrics for the MMA benchmark:

---

[1] https://www.cambridgeenglish.org/exams-and-tests/cefr/

**Ambiguity Accuracy (Amb_A)**  This metric is calculated as the percentage of questions where the model correctly answer for both paired images. A high Amb_A indicates that the model does not simply latch onto one possible interpretation of the ambiguity. Instead, it effectively integrates visual information from images to arrive at the most appropriate answer for each scenario. The examples are given in Figure 2.

## 4.3 Main results

Table 3: **Overall performance comparisons (Amb_A) of MLLMs on different ambiguity types**. The best results are **bold**. The second best results are underlined.

| | Adjective (30) | Noun (238) | Verb (16) | Attachment (24) | Coordination (46) | Structural (14) | Pragmatic (132) | Idiom (22) | Lexical (284) | Syntactic (84) | Semantic (154) | Overall (522) |
|---|---|---|---|---|---|---|---|---|---|---|---|---|
| **Proprietary MLLMs:** | | | | | | | | | | | | |
| GPT-4 Vision (OpenAI, 2024b) | **0.87** | 0.748 | 0.63 | 0.23 | 0.41 | 0.29 | 0.68 | 0.62 | 0.75 | 0.33 | 0.65 | 0.65 |
| GPT-4o-0513 (OpenAI, 2024c) | 0.80 | 0.82 | **0.88** | 0.08 | 0.41 | 0.43 | 0.65 | **0.73** | 0.82 | 0.31 | **0.69** | 0.70 |
| Gemini 1.5 Pro(DeeoMind, 2024) | 0.79 | 0.75 | 0.83 | 0.54 | **0.59** | 0.14 | **0.74** | 0.38 | 0.76 | 0.50 | 0.57 | 0.66 |
| Gemini 1.0 Pro Vision (DeeoMind, 2023) | 0.69 | 0.68 | 0.40 | 0.00 | 0.32 | 0.00 | 0.41 | 0.29 | 0.67 | 0.17 | 0.35 | 0.49 |
| Claude 3 Opus (Anthropic, 2024) | 0.73 | 0.56 | 0.38 | 0.00 | 0.16 | 0.00 | 0.25 | 0.16 | 0.57 | 0.08 | 0.21 | 0.38 |
| Claude 3.5 Sonnet (Anthropic, 2024) | 0.83 | **0.83** | 0.86 | **0.67** | 0.47 | 0.50 | 0.73 | 0.30 | **0.83** | 0.53 | 0.67 | **0.74** |
| GPT-4o-0806 (OpenAI, 2024c) | 0.80 | 0.82 | 0.75 | 0.31 | 0.55 | **0.57** | 0.73 | 0.65 | 0.81 | 0.48 | **0.69** | 0.72 |
| Proprietary Average | 0.79 | 0.75 | 0.67 | 0.26 | 0.41 | 0.28 | 0.60 | 0.45 | 0.75 | 0.34 | 0.55 | 0.62 |
| **Open-source MLLMs:** | | | | | | | | | | | | |
| LLaVA-NeXT-34B (Liu et al., 2024) | 0.87 | 0.80 | 0.5 | 0.08 | 0.59 | 0.00 | 0.40 | 0.41 | 0.79 | 0.33 | 0.40 | 0.60 |
| LLaVA-NeXT-13B (Liu et al., 2024) | 0.67 | 0.64 | 0.38 | 0 | 0.09 | 0 | 0.33 | 0.59 | 0.63 | 0.05 | 0.45 | 0.48 |
| LLaVA-NeXT-7B (Liu et al., 2024) | 0.13 | 0.60 | 0.13 | 0 | 0 | 0.14 | 0.28 | 0.27 | 0.52 | 0.02 | 0.27 | 0.37 |
| VILA1.5-40b (Lin et al., 2023) | 0.73 | 0.81 | 0.63 | 0.23 | 0.55 | 0.00 | 0.60 | 0.38 | 0.79 | 0.36 | 0.49 | 0.63 |
| VILA1.5-13b (Lin et al., 2023) | 0.40 | 0.70 | 0.13 | 0.00 | 0.14 | 0.14 | 0.38 | 0.49 | 0.63 | 0.10 | 0.43 | 0.49 |
| VILA1.5-3b (Lin et al., 2023) | 0.13 | 0.18 | 0.25 | 0.08 | 0.09 | 0.14 | 0.18 | 0.08 | 0.18 | 0.10 | 0.13 | 0.15 |
| Yi-VL-34b (AI et al., 2024) | 0.73 | 0.63 | 0.25 | 0.08 | 0.14 | 0.00 | 0.45 | 0.24 | 0.62 | 0.10 | 0.35 | 0.46 |
| InternVL-Chat-V1-5 (Chen et al., 2024b) | 0.80 | **0.83** | 0.63 | 0.38 | 0.55 | 0.14 | 0.70 | 0.54 | 0.82 | 0.43 | 0.62 | 0.70 |
| InternVL2-40B (Chen et al., 2024b) | 0.60 | 0.60 | 0.50 | 0.15 | **0.59** | 0.43 | 0.50 | 0.27 | 0.59 | 0.43 | 0.47 | 0.53 |
| Cogvlm2 (Wang et al., 2023) | 0.33 | 0.57 | 0.13 | 0.00 | 0.36 | 0.00 | 0.38 | 0.43 | 0.52 | 0.19 | 0.40 | 0.43 |
| DeepSeek-VL (Lu et al., 2024) | 0.47 | 0.70 | 0.50 | 0.23 | 0.27 | 0.00 | 0.53 | 0.38 | 0.66 | 0.21 | 0.45 | 0.53 |
| MiniCPM-Llama3-V 2.5 (OpenBMB, 2024) | 0.00 | 0.12 | 0.25 | 0.15 | 0.14 | 0.00 | 0.23 | 0.05 | 0.11 | 0.12 | 0.14 | 0.12 |
| HPT 1.5 Air (HYPERGAI, 2024) | 0.80 | 0.76 | 0.25 | 0.23 | 0.23 | 0.00 | 0.53 | 0.59 | 0.73 | 0.19 | 0.56 | 0.59 |
| Qwen2-VL-72B (Wang et al., 2024b) | 0.79 | 0.72 | 0.50 | 0.40 | 0.41 | 0.50 | 0.58 | 0.10 | 0.72 | 0.42 | 0.51 | 0.61 |
| Qwen2-VL-7B (Wang et al., 2024b) | 0.93 | 0.77 | 0.83 | 0.00 | 0.37 | 0.33 | 0.57 | 0.10 | 0.79 | 0.26 | 0.50 | 0.62 |
| LLaVA-OneVision-72B (Li et al., 2024) | 0.93 | 0.61 | 0.50 | 0.77 | 0.59 | 0.14 | 0.41 | 0.00 | 0.63 | 0.57 | 0.35 | 0.54 |
| LLaVA-OneVision-7B Li et al. (2024) | 0.47 | 0.74 | 0.38 | 0.23 | 0.45 | 0.00 | 0.50 | 0.18 | 0.69 | 0.31 | 0.45 | 0.56 |
| Open-sourced Average | 0.58 | 0.63 | 0.39 | 0.18 | 0.33 | 0.12 | 0.44 | 0.30 | 0.61 | 0.25 | 0.41 | 0.50 |
| **Human:** | | | | | | | | | | | | |
| Human Average | 0.83 | 0.93 | 0.83 | 1.00 | 0.90 | 0.63 | 0.82 | 0.98 | 0.91 | 0.89 | 0.85 | 0.89 |

### 4.3.1 Overall performance

As shown in Table 3, the mean ambiguity accuracy (Amb_A) of MLLMs varies significantly across different ambiguity types, highlighting challenges in handling structural and pragmatic ambiguities. However, a clear gap remains when comparing these models to human performance, which significantly outperforms the MLLMs.

Proprietary models, such as Claude 3.5 Sonnet (74%), achieve the best overall performance on Amb_A and excel at handling lexical ambiguities (83%). Among open-source models, InternVL-Chat-V1-5 (69.7%) shows strong performance, particularly in lexical categories (82%), achieving nearly comparable performance to Claude 3.5 Sonnet.

Despite these advancements, the best-performing models like Claude 3.5 Sonnet and GPT-4o still show a substantial gap when compared to human performance. Claude 3.5 Sonnet achieves an overall accuracy of 74%, which is 15% lower than the human benchmark of 89%. Similarly, GPT-4o performs 19% lower than human performance with an overall accuracy of 70%. Gemini-1.5 pro and InternVL-Chat-V1-5 also underperform humans by 23% and 19%, respectively, with overall accuracy of 66% and 70%. This significant performance gap is particularly evident in tasks involving syntactic and semantic ambiguities. For example, Claude 3.5 Sonnet and GPT-4o achieve accuracy of 53% and 31% in syntactic ambiguities, respectively, compared to the human accuracy of 89%. In semantic ambiguities, Claude 3.5 Sonnet and GPT-4o achieve 67% and 69%, respectively, while humans achieve 85%.

### 4.3.2 Explore the reasons for the gap between SOTA models and human

In order to investigate the reasons behind the performance gap between models and humans, we conducted the following experiments:

**MLLMs Performance with Text-Only Input:** Initially, we explored if the inherent **complexity of the tasks or human-crafted questions** might contribute to the performance gap. To this end, we assessed the accuracy of MLLMs when they were provided solely with text inputs. The metric

Table 4: **MLLM Performance with Text-Only Input**: We assessed the ratio of selecting one of the correct answers when MLLMs are given text-only input. This metric is used to measure the language understanding ability of MLLMs, addressing concerns about the potential bias introduced by human-crafted questions.

| Model | Attachment | Overall |
|---|---|---|
| Claude 3.5 Sonnet | 0.77 | 0.83 |
| GPT-4 Vision | 1.00 | 0.90 |
| Claude 3 Opus | 1.00 | 0.88 |
| GPT-4o-2024-05-13 | 0.85 | 0.89 |
| GPT-4o-2024-08-06 | 0.85 | 0.88 |
| InternVL-Chat-V1-5 | 0.85 | 0.86 |

Table 5: **MLLMs' Error Consistency Rate**: This metric represents the ratio of instances where MLLMs provide the same answer even when presented with two different images. It is used to measure the extent to which MLLMs neglect image information in clarifying ambiguities during the question-answering process.

| Model | Lexical | Syntactic | Semantic | Overall |
|---|---|---|---|---|
| Claude 3 Opus | 0.86 | 0.72 | 0.89 | 0.84 |
| GPT-4o-2024-05-13 | 0.72 | 0.83 | 0.79 | 0.78 |
| InternVL-Chat-V1-5 | 0.62 | 0.83 | 0.69 | 0.71 |
| DeepSeek-VL | 0.69 | 0.76 | 0.69 | 0.71 |
| HPT 1.5 Air | 0.66 | 0.82 | 0.74 | 0.74 |
| VILA1.5-40b | 0.73 | 0.78 | 0.95 | 0.83 |
| Yi-VL-34b | 0.65 | 0.84 | 0.86 | 0.77 |

used represents the rate at which the model's response matches one of the correct answers in each pair of data (ambiguity pair), it is considered accurate. As shown in Table 4, MLLMs demonstrate high accuracy when provided with only text input. The overall accuracy rates range from 83% to 90%, with GPT-4 Vision achieving the highest at 90%. Notably, performance is consistently strong across lexical, syntactic, and semantic categories, with most models scoring above 80% in each. Claude 3.5 Sonnet shows the most balanced performance across categories, while others like InternVL-Chat-V1-5 exhibit some variability (e.g., 90% lexical vs. 74% syntactic). These results indicate that minor textual issues have minimal impact on MLLMs' ability to select correct answers.

**MLLMs' Error Consistency Rate:** This Error Consistency Rate (ECR) - defined as the rate of selecting the same answer among incorrect cases. As shown in Table 5, when MLLMs made errors, they demonstrated a high consistency rate in choosing the same option twice. This rate ranged from 71% to 84% overall, depending on the model. The consistently high rates across lexical, syntactic, and semantic levels indicate that these models often failed to effectively leverage visual information when answering questions. Instead, they exhibited a strong bias towards the text modality, relying primarily on textual cues even when visual information was available. More error analysis are given in Appendix A.5.

In summary, the experimental results clearly indicate that the performance gap between MLLMs and humans does not stem from the inherent complexity of the tasks or the construction of the questions, as evidenced by the high accuracy rates with text-only inputs. Rather, the persistent performance gap is largely due to the models' failure to adequately process and integrate visual information to clarify the textual ambiguity. The tendency of MLLMs to repeat the same answers, even when presented with different visual contexts, highlights a pronounced bias towards textual information instead of leveraging visual information.

### 4.3.3 GAP BETWEEN PROPRIETARY MODELS AND OPEN-SOURCED MODELS

On average, proprietary models demonstrate better performance than open-sourced models in the MMA task. Specifically, proprietary models achieve 57.70% in Amb_A, while open-sourced models obtain 47.04% in Amb_A as Table 3 shows. For both indicators, proprietary models outperform open-sourced models.

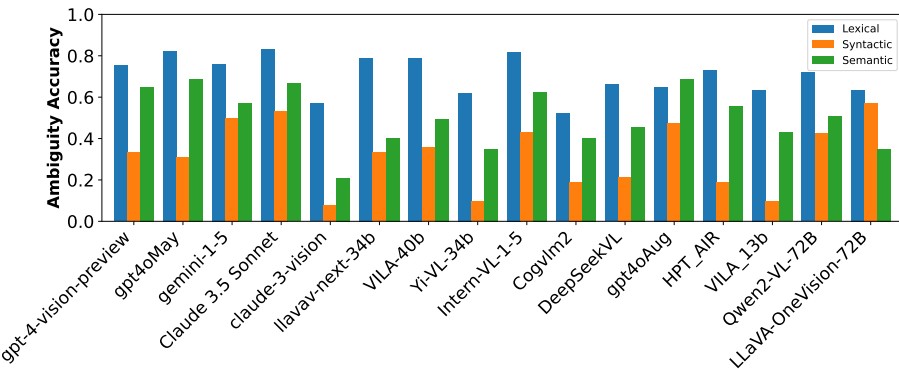

Figure 5: All models except MiniCPM-Llama3-V 2.5 perform better on Lexical ambiguity than Syntactic ambiguity and semantic ambiguity for Ambiguity Accuracy.

### 4.3.4 SYNTACTIC AMBIGUITY AND SEMANTIC AMBIGUITY ARE MORE CHALLENGING THAN LEXICAL AMBIGUITY

For both Amb_A, all models (except MiniCPM) perform better on lexical ambiguity and worse on syntactic and semantic ambiguities (Figure 5). Lexical ambiguity, which involves straightforward word meanings, is easier for models to handle. For example, InternVL-Chat-V1-5 achieves an accuracy of 82% on lexical ambiguities, significantly higher than its performance on syntactic (43%) and semantic (62%) ambiguities. This trend is consistent across most models; for instance, GPT-4o shows 82% accuracy on lexical ambiguities but drops to 31% and 69% on syntactic and semantic ambiguities, respectively.

Syntactic ambiguities present a unique challenge because they involve the relationships between components within a sentence. Often, even a short modifier can introduce ambiguity, making it difficult for models to resolve these cases without fine-grained analysis. To effectively handle syntactic ambiguities, models need not only a more granular approach to language processing but also the capability to accurately recognize positional relationships and details in images. This requires a higher level of precision compared to lexical and semantic ambiguities. Models like Cogvlm2 and VILA-3b, for example, perform poorly in this category, with accuracies of 19% and 10%, respectively.

Similarly, semantic ambiguities, which involve nuanced meanings and context, are also difficult for models to resolve. For instance, VILA-40b achieves only 49% accuracy on semantic ambiguities, despite a higher performance on lexical (79%).

### 4.3.5 SCALING LAW ON MMA

To comprehend whether the parameter number affects performance on the MMA benchmark, we conducted experiments on the same series of models with varying sizes, all trained on similar data. As Figure 4 shows, there is a clear improvement in ambiguity accuracy as the parameter count increases across different ambiguity types. For instance, larger models like VILA1.5-40B consistently outperform smaller ones such as VILA1.5-3B. The larger models show significant improvements in handling lexical, syntactic, and semantic ambiguities, demonstrating that increased parameters enhance the model's ability to understand and disambiguate complex, multi-modal inputs. This trend indicates a positive correlation between model size and performance on the MMA benchmark.

## 5 LIMITATION

**Data collection**    Due to constraints on the number of participants, the dataset size is limited in certain categories. Despite this limitation, we emphasize that the quality and representativeness of the dataset are more crucial for establishing a meaningful benchmark than merely the number of samples. As demonstrated in Table 3, the considerable performance discrepancy between human participants and MLLM responses underscores the benchmark's effectiveness in highlighting the current challenges that MLLMs face, particularly their inability to adequately utilize visual context to resolve textual ambiguities. Moving forward, we are committed to expanding the dataset in future iterations of the benchmark, aiming to broaden its scope and enhance its representational validity.

**Question design**  In our benchmark, both images and texts are designed to provide context information to model the multi-modal real-world cases. Due to the paper presentation problem, how to present some questions naturally presents certain challenges. We conducted experiments with text-only input and found that MLLMs demonstrate high accuracy, ranging from 83% to 90% (as Table 4 shows). However, when errors occurred, models consistently chose the same incorrect answers (as Table 5 shows). These results clearly indicate MLLMs have a strong bias towards text-based information and a failure to effectively incorporate visual context.

**Real-world likeness**  Some of images used in our benchmark are generated by generative models. The images in our benchmark are specifically chosen to provide the necessary context to clarify ambiguities in the accompanying texts. Due to the current limitations of search engines, which struggle with semantic search, it is challenging to find suitable images that naturally align with the required context (This doesn't mean that these images don't exist.). Therefore, using generated images is the most effective approach. They are instrumental in simulating the diverse and often unconventional situations that MLLMs encounter in real applications. MLLMs are expected to perform comparably to humans in these scenarios, regardless of the variability in inputs. However, our human study shows that humans can achieve approximately 90% accuracy on this benchmark without any additional interactions. This sharply contrasts with the average accuracies of 58% for closed-source models and 47% for open-source models.

## 6  FUTURE WORK

**Additional Modalities**  The world is multimodal rather than just bimodal. For instance, audio plays an important role in daily life, and there are some ambiguities caused by audio. For example, the phrases "He's a great **rapper**" and "He's a great **wrapper**" sound similar but refer to completely different things. With a concrete scene provided, the meaning of a segment of audio can be uniquely determined.

**Additional Languages**  Language-specific features and rhetorical devices vary widely, influencing how information is processed and understood. For instance, the use of 'Huwen' in ancient Chinese literature requires an understanding of how meanings are intricately split and reconnected across sentences. Expanding MLLMs to accommodate the linguistic structures and subtleties of various languages could improve their applicability and accuracy in global communication contexts. This development would necessitate models that are not only multilingual but also sensitive to cultural and contextual nuances within languages.

**Multiple Images per Sentence**  Lexical ambiguities can extend beyond dual interpretations, with some words or phrases having multiple meanings. Current models often limit context to one or two visual representations per sentence. By providing multiple images that correspond to each potential meaning of a sentence, MLLMs can be trained to discern finer distinctions in word usage and context. This enhancement would allow models to handle more complex scenarios where multiple interpretations are valid, reflecting the true complexity of human language and cognition.

## 7  CONCLUSION

This paper introduces MMA, the first benchmark designed specifically to evaluate the ability of Multimodal Large Language Models (MLLMs) to understand and respond to ambiguous queries. MMA leverages a multiple-choice visual question-answering format, presenting MLLMs with a question and two images depicting contrasting scenarios that lead to different correct answers. Our evaluation of 16 MLLMs, including both limited-access and open-sourced models, reveals a significant performance gap compared to human performance. While humans achieve an accuracy of 88.97%, the MLLMs average only 50.59% accuracy. This indicates a fundamental challenge for current MLLMs: effectively integrating scenario-specific visual information to disambiguate questions and arrive at the correct answer. Even the top-performing model, GPT-4o and Claude3.5-Sonnet, attains only about 70.00% accuracy, highlighting considerable room for developing MLLMs that can effectively leverage visual information to clarify the textual ambiguity and capable of human-level understanding and reasoning in complex, real-world scenarios.

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

# A APPENDIX

## A.1 DISTRIBUTION OF DATASET

As shown in Figure 6, the MMA dataset consists of 522 images and 261 questions, covering three main types of ambiguity: lexical ambiguity, syntactic ambiguity, and semantic ambiguity. These main categories are further divided into eight sub-categories: noun ambiguity, verb ambiguity, and adjective ambiguity (under lexical ambiguity); attachment ambiguity, coordination ambiguity, and structural ambiguity (under syntactic ambiguity); and pragmatic ambiguity and idiomatic ambiguity (under semantic ambiguity).

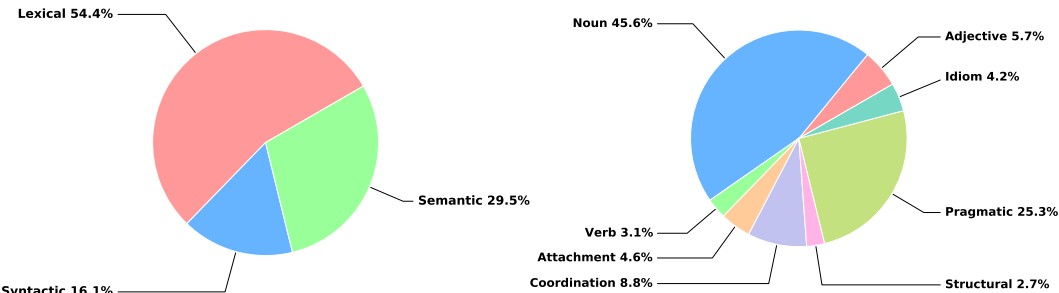

Figure 6: Ambiguity Type Composition of MMA benchmark

## A.2 BENCHMARK AND EVALUATION RESOURCES

To facilitate benchmarking, we've made the dataset available.

For evaluation purposes, you can utilize the code provided in our github webpage.

## A.3 IMAGE USAGE AND COPYRIGHT CLAIMS

Our images are sourced from search engines (such as Google and Bing) and text-to-image models (such as Stable-Diffusion and DALL-E). All collected images are used exclusively to support our non-profit research project, MMA Benchmark. If you own the copyright to any images used in this project and believe that your rights have been violated, please contact us. We are willing to compensate for the usage of your images.

## A.4 ABLATION STUDY

**Same images with lexical or semantic questions** To understand why MLLMs perform better on lexical ambiguity compared to semantic ambiguity, we explored how changing the question type on noun ambiguity impacts their performance. We created two versions of questions for noun categories: the first being the most direct, "What's the meaning of <Noun>?", and the second incorporating reasoning into the question. For example, given an image of a table, a synonym question for lexical ambiguity might be "What is the meaning of table?" where the model identifies "table" as a piece of furniture. In contrast, a reasoning question for semantic ambiguity would be "How can we best utilize the space on this table?" which requires the model to consider various uses of the table. This type of question tests the model's ability to perform object grounding and higher-order reasoning, areas where MLLMs often show weaker performance due to their reliance on pattern recognition rather than true comprehension. More examples are given in Appendix.

As Figure 7 shows, GPT-4 Vision performs well on noun word ambiguity with a score of 90% but drops to 59% on noun reasoning ambiguity. Similarly, Gemini-1.5 shows a significant drop from 83% in noun word ambiguity to 63% in noun reasoning ambiguity. Intern-VL-Chat-V1-5, while achieving 92% in noun word ambiguity, sees a decline to 75% in noun reasoning ambiguity. These examples highlight the challenges MLLMs face in understanding and reasoning about more complex and context-dependent scenarios.

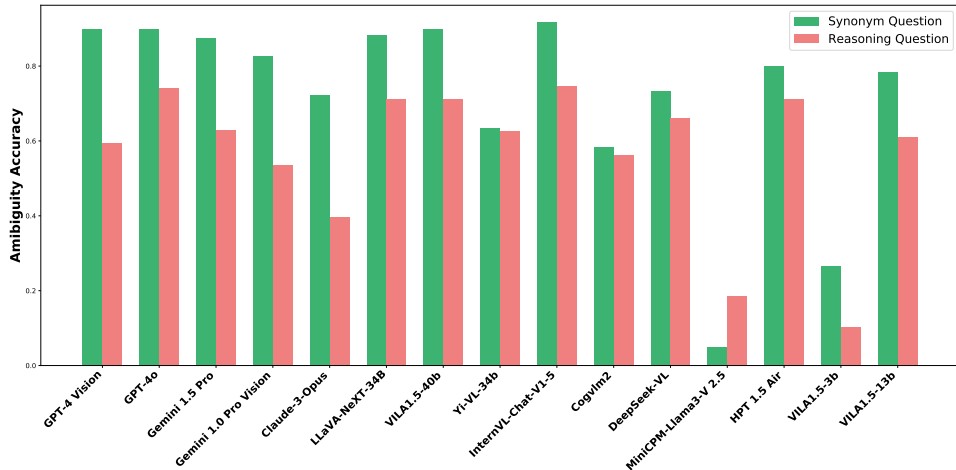

Figure 7: **The performance comparison for question types**, where The Noun_word refers to questions that solely inquire about the meaning of a noun word, while the Noun_reasoning involves questions that require the reasoning ability to answer. The details and examples are given in Appendix.

## A.5 ERROR ANALYSIS

Errors can be categorized into three main types: **uni-modal image issues, uni-modal text issues, and cross-modal text bias.** An analysis of the error distribution in GPT-4o reveals that cross-modal text bias errors constitute the majority of all errors(see Figure 8). This finding suggests that there is significant room for improvement MMA benchmark.

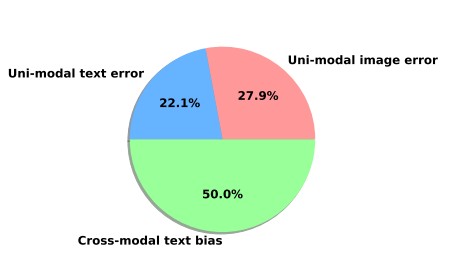

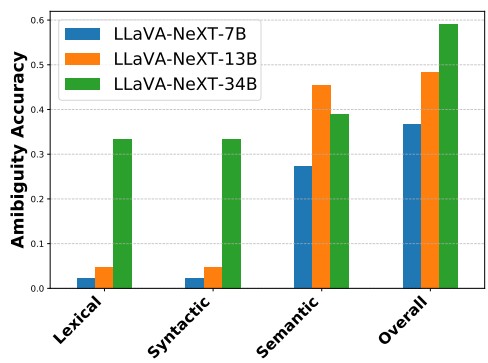

Figure 8: **Error type distribution of GPT-4o**, where we see cross-model text bias accounts for half of the cases.

Figure 9: The ablation study about the parameter number and the ambiguity accuracy performance on different ambiguity types.

**Uni-modal Image Issues (22.1%)**   In this type of error, the model fails to capture the essential information conveyed by the image. To address this issue, visual prompts, such as red bounding boxes, can be incorporated to redistribute the attention of the Multimodal Large Language Model (MLLM). By emphasizing the crucial elements of the image, the model can be guided towards generating the correct answer based on the key visual information(see Figure 10).

**Uni-modal Text Issues (27.9%)**   In this type of error, the model successfully captures the essential information from the image but provides an incorrect answer due to misinterpreting the text options. To resolve this issue, text prompts can be introduced to guide the MLLMs towards a proper understanding of the textual content. By ensuring accurate comprehension of the text, these prompts can help the model arrive at the correct answer (see Figure 11).

**Cross-modal Text Bias (50.0%)**   In this category of errors, the model successfully identifies the essential information in the image and comprehends the text options. However, it provides an incorrect answer due to overlooking certain aspects of the visual information while overemphasizing

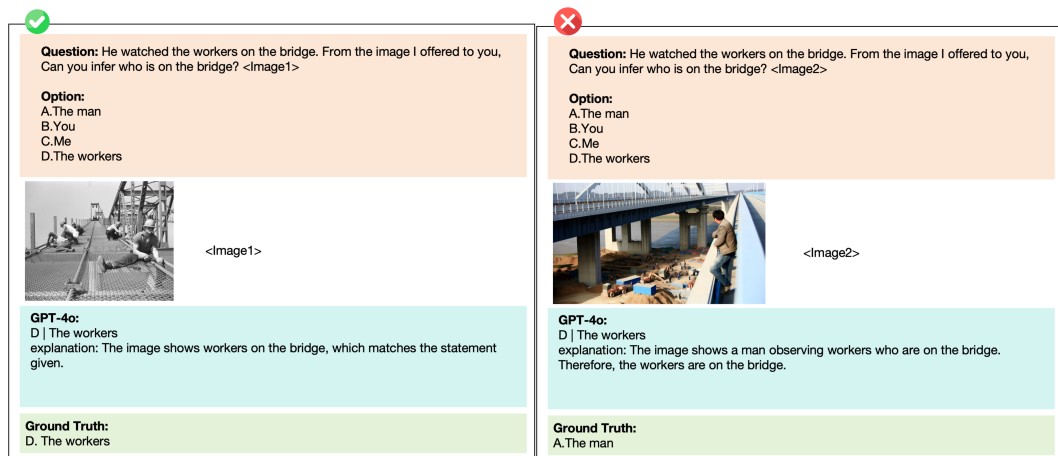

Figure 10: Uni-modal Image Issues: the model fails to capture the essential information conveyed by the image.

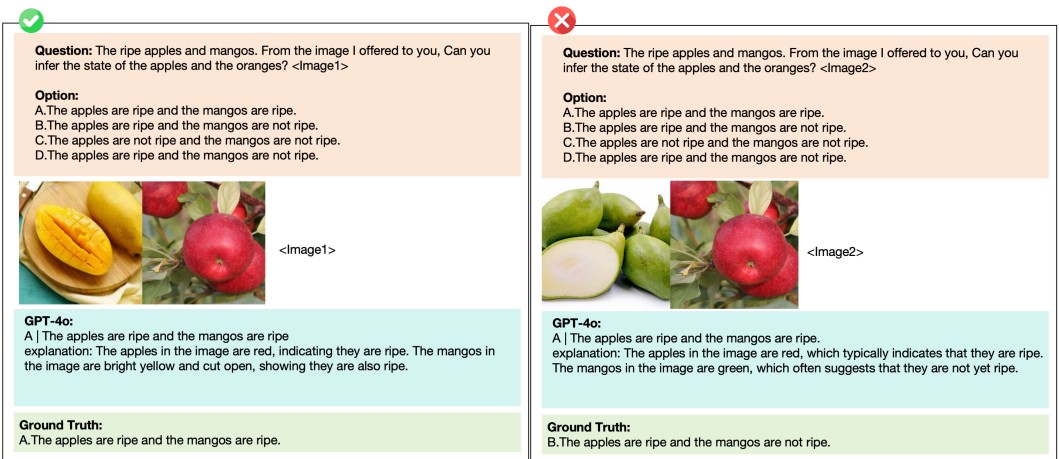

Figure 11: Uni-modal Text Issues: the model successfully captures the essential information from the image but provides an incorrect answer due to misinterpreting the text options.

the textual information. To mitigate this issue, text prompts can be introduced to rebalance the attention between the image and text modalities(see Figure 12). By adjusting the relative importance of visual and textual cues, the model can be encouraged to arrive at the correct answer by considering all relevant information from both modalities.

## A.6 HUMAN EVALUATION

To validate our dataset and assess the performance difference between humans and models, we invited five people to participate in benchmark testing. As shown in the table, for each sub-ambiguity class, at least one person achieves an ambiguity accuracy of over 90%, with the exception of Pragmatic ambiguity, where the highest accuracy is 88%. These results demonstrate that our dataset is well-

|  | Adjective (30) | Noun (238) | Verb (16) | Attachment (24) | Coordination (46) | Structural (14) | Pragmatic (132) | Idiom (22) | Lexical (284) | Syntactic (84) | Semantic (154) | Overall (522) |
|---|---|---|---|---|---|---|---|---|---|---|---|---|
| Person1 | 0.60 | 0.88 | 0.88 | 1.00 | 0.77 | 0.00 | 0.74 | 0.91 | 0.85 | 0.71 | 0.77 | 0.80 |
| Person2 | 0.93 | 0.97 | 1.00 | 1.00 | 0.86 | 1.00 | 0.83 | 1.00 | 0.96 | 0.93 | 0.86 | 0.93 |
| Person3 | 0.80 | 0.94 | 0.50 | 1.00 | 0.91 | 0.71 | 0.88 | 1.00 | 0.90 | 0.90 | 0.90 | 0.90 |
| Person4 | 0.93 | 0.93 | 1.00 | 1.00 | 0.95 | 0.71 | 0.85 | 1.00 | 0.94 | 0.93 | 0.87 | 0.92 |
| Person5 | 0.87 | 0.93 | 0.75 | 1.00 | 1.00 | 0.71 | 0.82 | 1.00 | 0.92 | 0.95 | 0.84 | 0.90 |

Table 6: Five people have different performance across different types of ambiguities

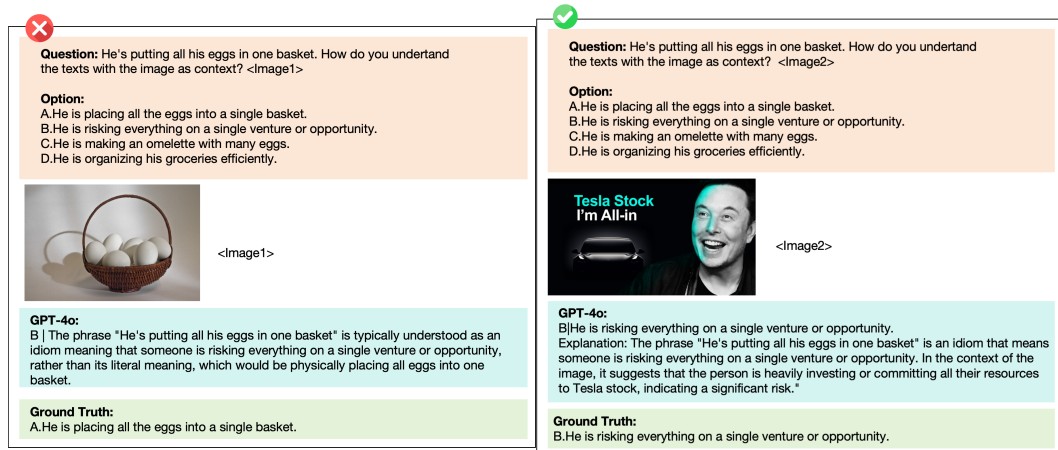

Figure 12: Cross-modal Text Bias: the model successfully captures the essential information from the image but provides an incorrect answer due to misinterpreting the text options.

constructed and solvable by humans, serving as a strong validation of the dataset's quality and the feasibility of the task. Humans may fail to answer questions correctly due to a lack of knowledge (such as not understanding the meaning of an idiom), being confused by misleading or similar answer options with subtle differences, or struggling to correlate images with text (particularly when the text contains advanced expressions or extended meanings). Here is an example where all respondents have failed to provide the correct answer Figure 13.

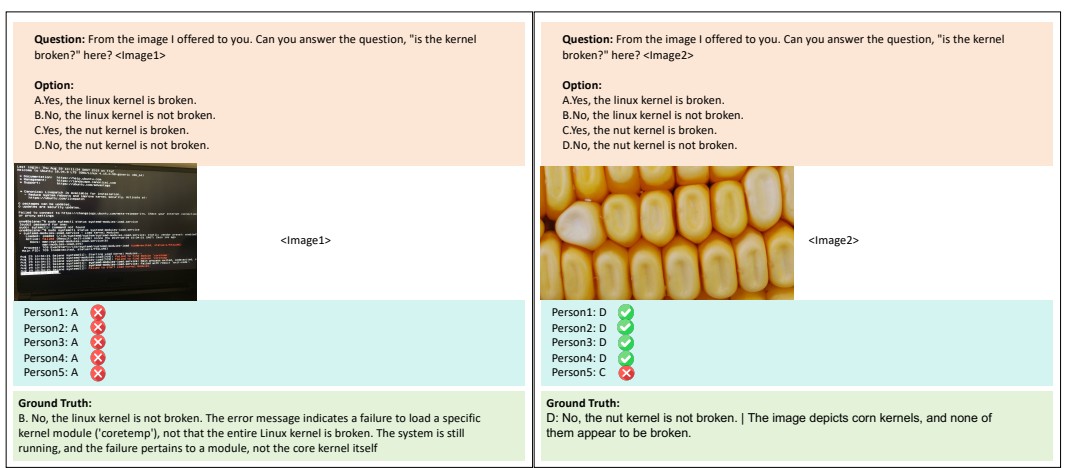

Figure 13: Case: All respondents make wrong choices

### A.7 SCALING LAW WITH LLAVA SERIES MODELS

As for Llava series models, the scaling law holds true for most metrics, with performance continuously improving as the model's parameter count increases(see Figure 9). This provides further evidence for the scaling law on the MMA benchmark. However, there is an exception when it comes to semantic ambiguity, where the middle-sized model performs best.

### A.8 CASE STUDY

More examples of GPT-4o will be presented in this section.

**Example of GPT-4o**

1. Example of Coordination Ambiguity

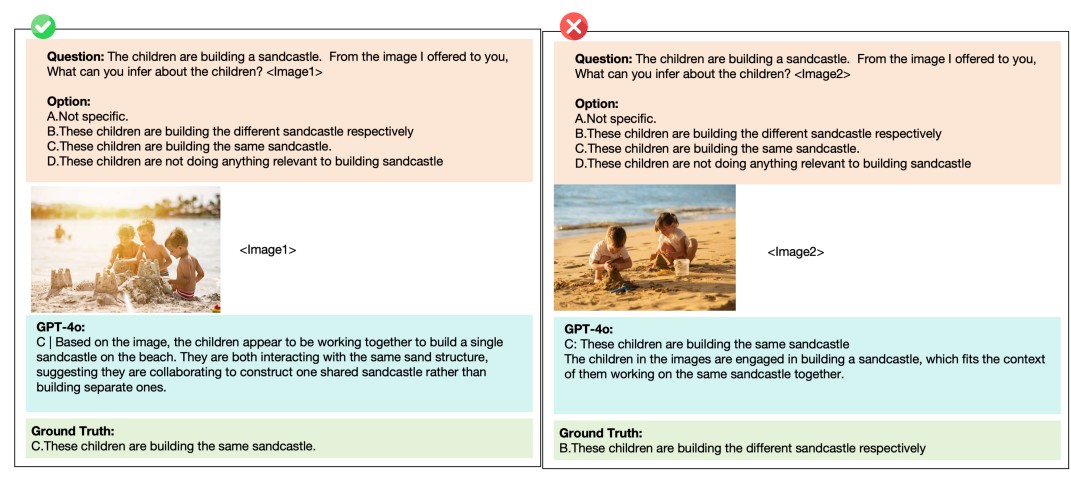

Figure 14: Example of Coordination Ambiguity

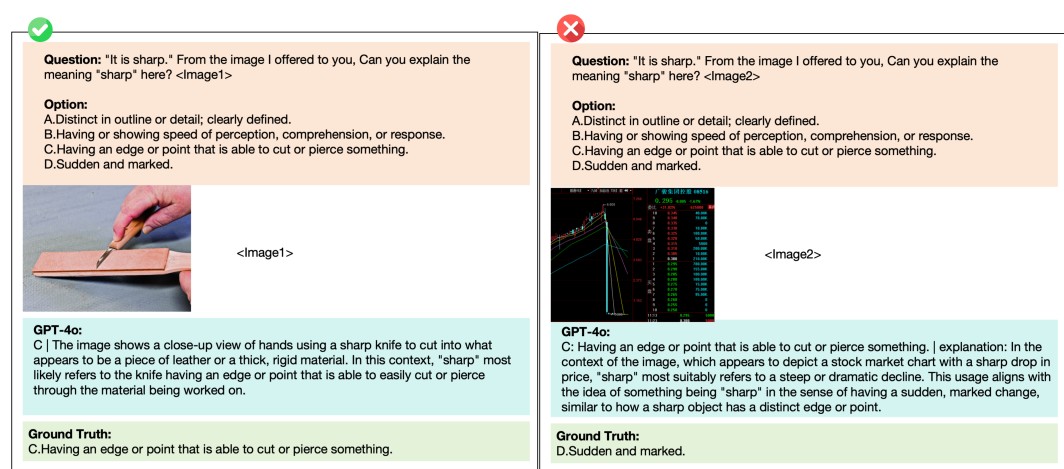

Figure 15: Example of Adjective Ambiguity

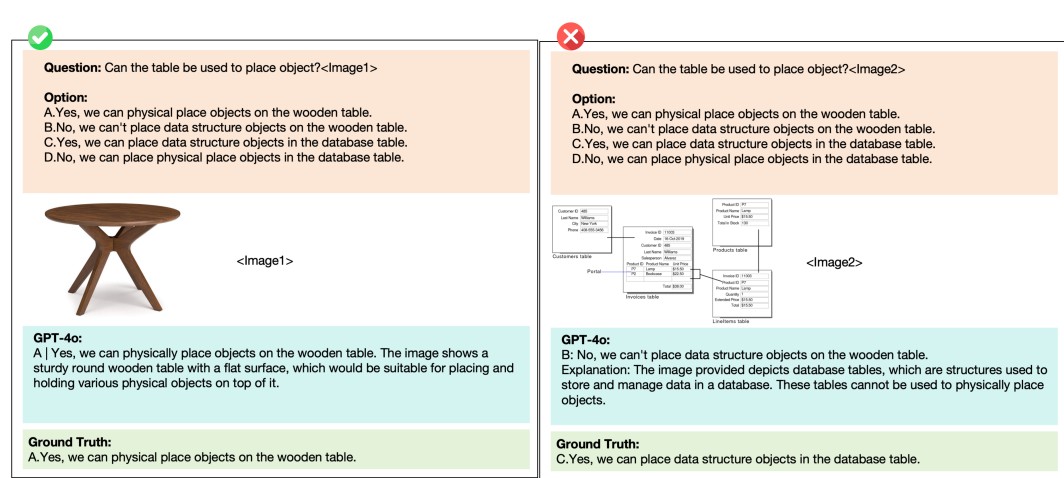

Figure 16: Example of Noun Ambiguity

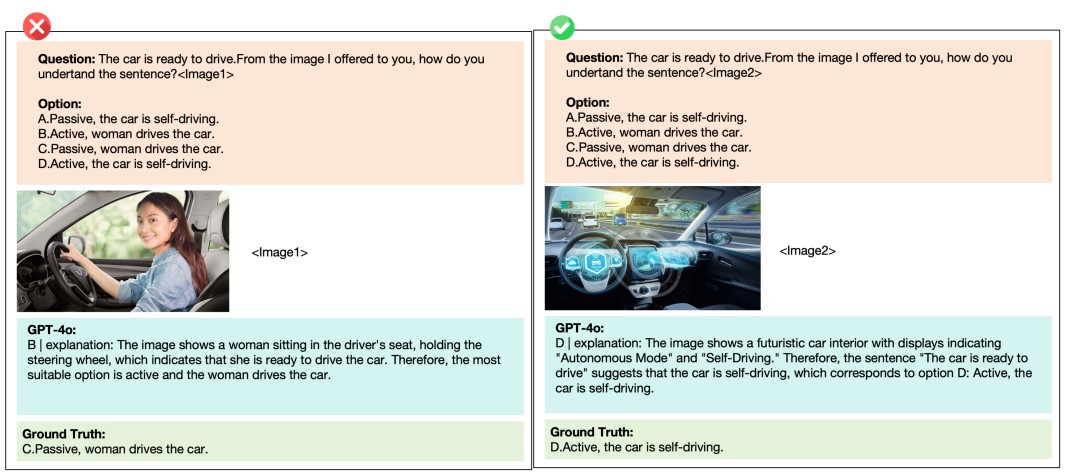

Figure 17: Example of Structural Ambiguity

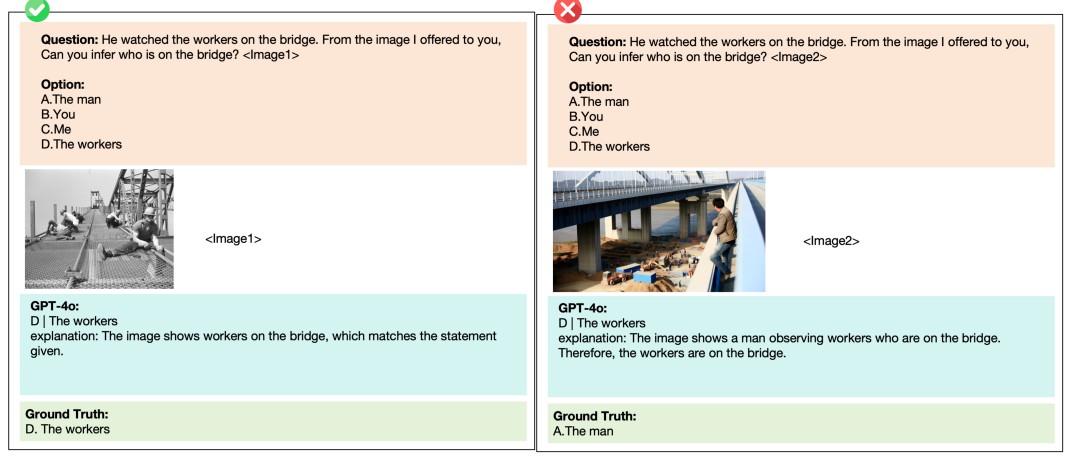

Figure 18: Example of Attachment Ambiguity

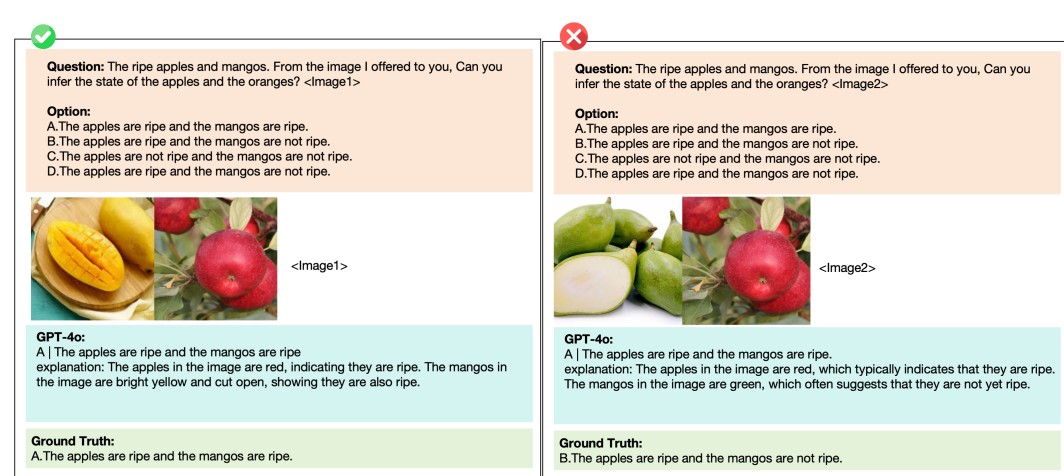

Figure 19: Example of Coordination Ambiguity

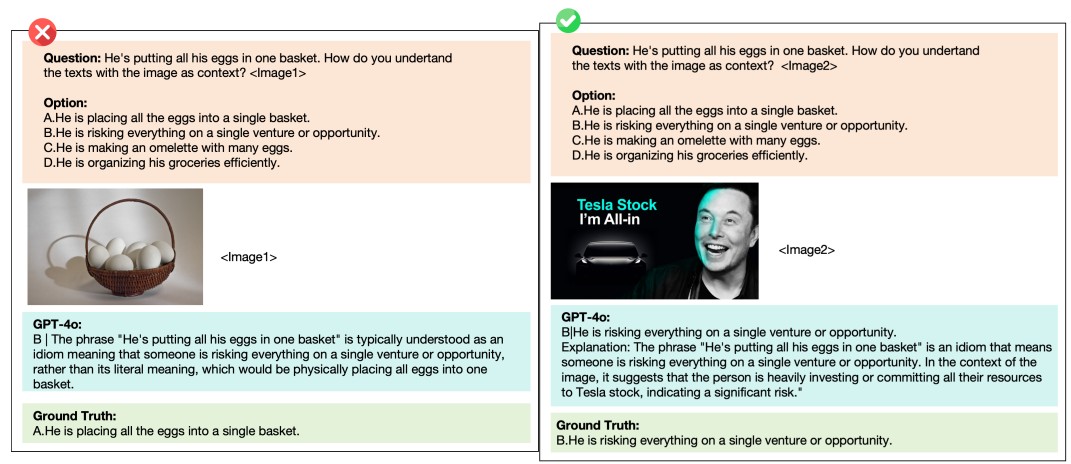

Figure 20: Example of Idiomatic Ambiguity

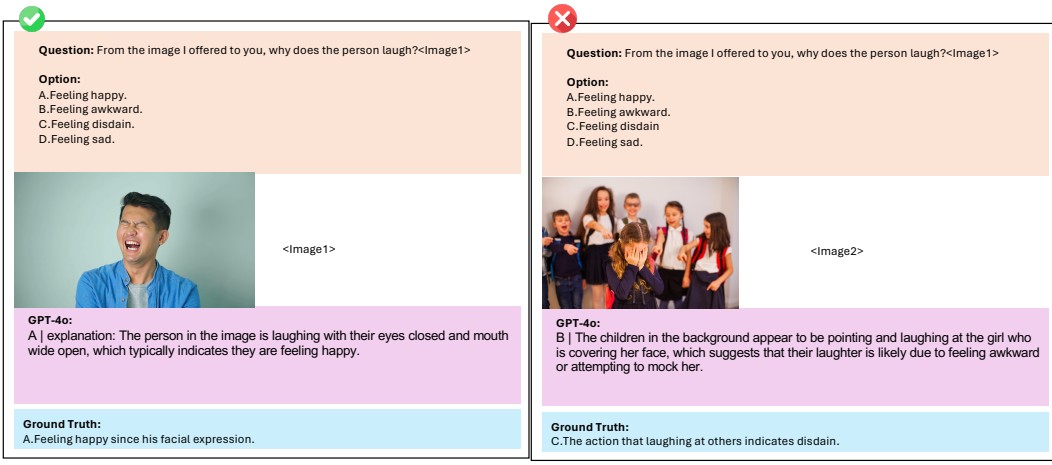

Figure 21: Example of Verb Ambiguity

