# OpenReview forum: "MMA: Benchmarking Multi-Modal Large Language Model in Ambiguity Contexts"
_ICLR.cc/2025/Conference — ICLR 2025 Conference Withdrawn Submission_

### Official Review · Reviewer_rr48 · 2024-10-26

**Soundness:** 3
**Presentation:** 3
**Contribution:** 3
**Rating:** 5
**Confidence:** 4

**Summary:**

This paper introduces a benchmark for evaluating multimodal large language models’ (MLLMs) understanding of ambiguous language. The benchmark contains 261 questions and 522 natural or generated images covering lexical, syntactic, and semantic ambiguities. The authors evaluated 16 state-of-the-art MLLMs and found significant performance gaps between MLLMs and human accuracy. While MLLMs handle lexical ambiguities relatively well, they struggle with other types. Detailed evaluation using text-only data reveals that this performance gap is primarily due to the models’ limited ability to effectively integrate multimodal information.

**Strengths:**

- This paper provides a novel evaluation benchmark and it is an interesting addition to the existing multimodal evaluation.
- The benchmark is well-designed, covering multiple types of ambiguity.
- The paper is well-written, and the evaluation is thorough, covering both closed-source and open-source MLLMs, as well as human evaluation baselines.

**Weaknesses:**

- The proposed MMA dataset is relatively small.
- Some technical details are missing in the current paper. For example, how many people contributed to the dataset/question creations, annotation process/instructions, percentage of natural images versus generated images etc.
- Some technical practices:
  * The benchmark includes natural images with unknown licensing and are not under Creative Commons. Although the authors express a willingness to pay for these images in the appendix, this practice raises concerns.
  * Near-native English speakers are recruited for human evaluations instead native English speakers, such design choice is not well justified.

**Questions:**

Question:
- Why use near-native English speakers for evaluations?

Comments:
- The paper could potentially benefit from deeper analysis, such as:
  * Currently scaling law is only shown of VILA1.5, it would be interesting to evaluate on other models with multiple sizes QWen, LLaVA-NeXT etc. If the scaling law exists for single modality (text) eval.
  * Quantitative and qualitative error analysis for MLLMs’ mistakes with natural images versus generated images.
  * A comparative analysis of human errors and mistakes made by MLLMs.

---

### Official Review · Reviewer_LHEX · 2024-11-02

**Soundness:** 3
**Presentation:** 2
**Contribution:** 2
**Rating:** 5
**Confidence:** 3

**Summary:**

The paper "MMA: Benchmarking Multi-Modal Large Language Models in Ambiguity Contexts" introduces the MMA benchmark, specifically designed to assess how Multi-Modal Large Language Models (MLLMs) handle ambiguous contexts. MMA presents questions associated with pairs of images that imply distinct scenarios, testing MLLMs' ability to resolve ambiguities through visual cues. The benchmark categorizes ambiguities into lexical, syntactic, and semantic types and evaluates 24 MLLMs, finding that while humans achieve an accuracy of nearly 89%, MLLMs struggle, achieving only around 53.22%. The results reveal that MLLMs are particularly challenged by syntactic ambiguities, with open-source models generally performing worse than proprietary ones. This study highlights areas for improvement in MLLMs’ integration of visual and textual information to handle ambiguity effectively.

**Strengths:**

Novel Benchmark: The introduction of MMA is innovative, providing a unique tool to evaluate MLLMs' handling of ambiguity across multiple categories.
Comprehensive Analysis: The study rigorously assesses performance across lexical, syntactic, and semantic ambiguity types, offering granular insights into model capabilities and limitations.
Valuable Findings for Future Improvements: The paper's results highlight weaknesses in current MLLMs, focusing on the need to improve model handling of syntactic and semantic ambiguities.
Broad Applicability: The benchmark has the potential for wider application in fields requiring high-precision understanding of ambiguous language, such as natural language processing in complex human-computer interactions.

**Weaknesses:**

The authors have placed the dataset distribution and construction details in the Appendix rather than the main text, which makes it challenging to follow the methodology while reading. Including these aspects in the main body of the paper would improve readability and help readers better understand the dataset's structure and rationale as they go through the paper.

**Questions:**

Data Curation: In the Appendix, I found a section on dataset distribution, but there seems to be no dedicated section on data curation. Based on A.3, the data is generated using text-to-image models. This raises questions about quality control, model output risk management, and the generated images' accuracy and safety. It would be beneficial to clarify how the outputs from these models are vetted for correctness and potential risks.

Dataset Construction: What is the source and motivation behind generating the specific types of questions? The rationale for distributing lexical, syntactic, and semantic ambiguities is also unclear. Why were these particular proportions chosen? If all three types are equally significant, a balanced distribution might be expected, or there should be an explanation if certain ambiguity types are more prevalent in real-life scenarios. (If I missed an explanation in the paper, please direct me to it.)

Table 4: I am unclear about the motivation behind this table. For instance, in the lexical category, items should ideally have two distinct interpretations. Correctly answering one interpretation does not imply the model has accurately handled the ambiguity by answering the other interpretation. Could you elaborate on the intended purpose of this analysis?

---

### Official Review · Reviewer_uqh8 · 2024-11-03

**Soundness:** 2
**Presentation:** 2
**Contribution:** 2
**Rating:** 3
**Confidence:** 4

**Summary:**

This paper presents MMA, a benchmark for evaluating Multi-Modal Large Language Models (MLLMs) in ambiguous contexts. It uses a multiple-choice visual question-answering format with 261 questions and associated pairs of images representing different scenarios. The benchmark categorizes ambiguities into lexical, syntactic, and semantic types. Through experiments on 24 MLLMs, it shows that models often overlook image information, perform better on lexical ambiguity, and that proprietary models generally outperform open-source ones.

**Strengths:**

1. This paper proposed a novel benchmark aimed at evaluating MLLMs’ ability to leverage visual information to clarify the ambiguities in texts. The task is designed to rely on both text and image information.
2. This work conducted comprehensive evaluations on 24 proprietary and open-sourced MLLMs. The categorization of ambiguities into different types allows for a detailed analysis of model performance.
3. The results, such as models' error consistency rate and the performance differences between different types of ambiguity and model types, offer valuable insights for future research and development.

**Weaknesses:**

1. The dataset size is limited in some categories due to constraints on the number of participants. This may affect the representativeness and generalizability of the results.
2. The authors should explore how to do the data collections in an autonomous way instead of human labors. Only on this manner, the data size can be increased significantly.
3. The authors should propose valuation suggestions for the MLLM data preparation, pretraining and posttraining. Some experiments on the open source MLLMs will add some value to this work.

**Questions:**

1. How can the dataset be expanded and improved to overcome the limitations of size and representativeness?
2. Do you have any suggestions for better MLLMs based on the findings you have on this dataset.

---

### Official Review · Reviewer_noQq · 2024-11-12

**Soundness:** 3
**Presentation:** 2
**Contribution:** 3
**Rating:** 5
**Confidence:** 4

**Summary:**

This paper introduces a designed to evaluate how well MLLMs handle ambiguity in language when provided with visual context. The main distinguished design is to provide the same text with two different visual contexts, and count the models correctly answer only if both answers are correct.

The benchmark consists of 261 (?) questions in total, each paired with two different images that suggest different interpretations of the same ambiguous text, requiring models to leverage visual information to disambiguate meaning. The questions are categorized into three types of ambiguity: lexical, syntactic, and semantic. The authors evaluate various MLLMs, including both proprietary and open-source models, on their ability to correctly interpret ambiguous questions when given different visual contexts.

**Strengths:**

* The benchmark's design of using paired images for the same ambiguous text is innovative and well-suited to the research question. Such a well-designed multi-image QA setting could be also useful to explore other capabilities of the MLLMs. Moreover, I think it could also prompt the community to think about how the existing multimodal understanding LLMs actually work, e.g., do they really understand the contexts?

* The categorization of ambiguity types (lexical, syntactic, semantic) provides a structured framework for analysis
* The human evaluation provides a strong baseline for comparison.

**Weaknesses:**

* The dataset size is relatively small (261 questions), particularly for certain ambiguity subcategories.
* Missing the annotation protocol of human baseline. The authors should provide a thorough description since it's the basis of almost all the analysis.

- The image data collection pipeline has potential ethical and biased issues : 1) use of generated images for some test cases might not fully reflect real-world scenarios (which could not be fully solved and also discussed in the literature); 2) the statement on the usage of images from "Google" is quite vague and could cause license issues.
- The multiple-choice format, while practical for evaluation, might not capture the full range of possible interpretations, especially considering the options could provide additional contexts for the models. How do you ensure that the multiple-choice options don't inadvertently provide hints about the correct interpretation? Could you elaborate on the option design process?

**Questions:**

1. Requiring clarification on the dataset quantity: in abstract and Appendix, the authors state there are 261 questions, and in Table 3 the numbers in parentheses are inconsistent.
2. How might the benchmark be adapted to evaluate MLLMs' ability to explain their reasoning process in addition to selecting the correct answer?
3. Could you provide more detail about the criteria used to determine whether generated images were suitable replacements for real-world images?

**Details Of Ethics Concerns:**

The author mention in the paper that images are collected from search engine and generated by models. While the former might cause potential licensing issues, the author do not provide and license and ethic statements in terms of the image using.

---

### Author Response · Authors · 2024-12-03
**General Response**

Thank you to all reviewers for acknowledging the novelty and interest of our benchmark. We appreciate your feedback and provide the following responses to address your concerns:

 **Q1:** The dataset size is relatively small, particularly for certain ambiguity subcategories.

A：We acknowledge that the limited dataset size could be seen as a potential drawback of our benchmark, but we respectfully believe that it does not significantly impact our main contributions.

1. **Highlighting Potential Improvements for MLLMs:** Our benchmark is the first to systematically evaluate the ability of MLLMs to leverage visual information to clarify language ambiguities. The comprehensive evaluation clearly demonstrates the significant performance gap between MLLMs and humans (e.g., 3.5 Sonnet achieving 74% vs. human 89%, MLLMs average 53% vs human 89% ) in leveraging multi-modal information to resolve ambiguities. **Identifying these areas for improvement is a critical step forward in advancing the field and can inspire more subsequent research efforts.**
2. **Data Representativeness as a Key Factor:** We acknowledge the concern regarding the limited dataset size in certain categories. However, we believe that the representativeness of the data is the key to a meaningful benchmark. Even with a smaller sample size, a benchmark can still serve as a robust standard if the data is well-representative. For example, MM-Vet [1] with only around 200 test samples, remains a widely recognized evaluation benchmark.  Before GPT-4o and Claude 3.5, no other MLLMs had achieved over 70% accuracy on the MM-Vet benchmark, **even 3 quarters after its release**. This underscores the value of well-crafted benchmarks, regardless of dataset size, in driving progress and setting high standards for model performance. In the future, we plan to expand our benchmark, similar to MM-Vet v2, to continue fairly evaluating future state-of-the-art MLLMs.

**Reference**
[1] Weihao Yu, et al. MM-Vet: Evaluating Large Multimodal Models for Integrated Capabilities. ICML 2024.

 **Q2:** Missing the annotation protocol of human baseline. The authors should provide a thorough description since it's the basis of almost all the analysis.

We acknowledge the importance of this protocol as the foundation for our analysis and provide the following detailed explanation:

1.	Annotator Recruitment:
We recruited five annotators with native or near-native proficiency in English, verified at the CEFR C1 level or above. This ensures a strong command of linguistic nuances, including those relevant to resolving textual ambiguities.
2.	Task Overview:
Annotators were presented with the complete MMA benchmark, which includes 261 multiple-choice questions, each paired with two images depicting different contexts.For each question, annotators were asked to select the most contextually appropriate answer based on the visual and textual information provided.
3.	Guidelines Provided to Annotators:
Annotators were given clear instructions emphasizing the need to integrate both visual and textual contexts to resolve ambiguities.They were explicitly instructed to avoid over-relying on either modality and to ensure their answers were consistent with the intended interpretation of the ambiguities.
4.	Evaluation Process:
Annotators independently completed the task without collaboration or external assistance.
Responses were recorded in a standardized answer sheet, ensuring consistency across all participants.


 **Q3:** Table 4: I am unclear about the motivation behind this table. For instance, in the lexical category, items should ideally have two distinct interpretations. Correctly answering one interpretation does not imply the model has accurately handled the ambiguity by answering the other interpretation. Could you elaborate on the intended purpose of this analysis?

A: The text-only evaluation was conducted for evaluating potential bias from human-crafted questions. By examining performance on text-only input, we aimed to investigate whether the performance gap observed in multimodal settings could be attributed to the inherent complexity of the tasks or biases introduced by human-crafted questions. High accuracy in text-only tasks suggests that the textual component alone does not hinder performance.

---

### Note · Authors · 2024-12-03

I have read and agree with the venue's withdrawal policy on behalf of myself and my co-authors.